# Benchmarking Farm Animal Welfare—A Novel Tool for Cross-Country Comparison Applied to Pig Production and Pork Consumption

**DOI:** 10.3390/ani10060955

**Published:** 2020-05-31

**Authors:** Peter Sandøe, Henning Otte Hansen, Helle Lottrup Halkjær Rhode, Hans Houe, Clare Palmer, Björn Forkman, Tove Christensen

**Affiliations:** 1Department of Food and Resource Economics, University of Copenhagen, Copenhagen, 1958 Frederiksberg C, Denmark; hoh@ifro.ku.dk (H.O.H.); tove@ifro.ku.dk (T.C.); 2Department of Veterinary and Animal Sciences, University of Copenhagen, Copenhagen, 1870 Frederiksberg C, Denmark; hek@sund.ku.dk (H.L.H.R.); houe@sund.ku.dk (H.H.); bjf@sund.ku.dk (B.F.); 3Department of Philosophy, Texas A&M University, College Station, TX 4237, USA; c.palmer@tamu.edu

**Keywords:** animal welfare, animal welfare assessment, animal welfare legislation, market-driven animal welfare, pig production, pork

## Abstract

**Simple Summary:**

In many countries, levels of animal welfare are driven by both national legislation and animal welfare labels. In order to assess what is being achieved at the national level, we developed a Benchmarking tool that combined information about legislation, welfare labels, market-shares, and expected welfare consequences. This tool was presented here and applied to pig production and pork consumption in five European countries. To assess how well a country is faring in terms of pig welfare, it was important not only to look at national pork production but also at the sourcing of pork by retailers and consumers since not all pork consumed in a country is produced domestically. Of the five countries studied, the three that have a large, export-oriented pig production industry—the Netherlands, Germany, and Denmark—had lower welfare levels in their production than the two—UK and Sweden—that had smaller pig production, mainly serving the home market. However, since the UK and Sweden also imported pork for retail and consumption from countries with lower levels of welfare, the contrast in welfare levels was smaller in consumption than in production.

**Abstract:**

A pluralist approach to farm animal welfare, combining animal welfare legislation with market-driven initiatives, has developed in many countries. To enable cross-country comparisons of pig welfare, a number of welfare dimensions, covering the features typically modified in legislative and market-driven welfare initiatives aimed at pig production, were defined. Five academic welfare experts valued the different welfare states within each dimension on a 0–10 scale, then assessed the relative contribution of each dimension to overall welfare on a 1–5 scale. By combining these values and weights with an inventory of pig welfare initiatives in five countries, the additional welfare generated by each initiative was calculated. Together with information on the national coverage of each initiative, the Benchmark value for each country’s production and consumption of pork could be calculated on a scale from 0 to 100. Two (Sweden and the UK) had a much higher Benchmark value than the rest. However, there was a drop in the Benchmark for consumption in Sweden and the UK (indicating imports from countries with lower-Benchmark values for production). Even though the experts differed in the values and weights ascribed to different initiatives, they were largely in agreement in their ranking of the countries.

## 1. Introduction

This paper presented a novel tool for comparing farm animal welfare achievement across countries. The key elements of the tool are (1) that it is based on a reasonably comprehensive list of parameters organized within a number of dimensions of welfare that influence the welfare of the farm animals in question; in this case, pigs; (2) that it uses expert opinion to value and weigh the effect of each of these parameters on the welfare of the animals; (3) that for each welfare initiative—for example, a set of legal requirements or requirements for a labeled product—Benchmark values are calculated on a scale from 0 to 100 that indicate the potential animal welfare outcome of the initiative, and, finally, (4) that by combining that information with information on what proportion of the nation’s production or consumption falls under each initiative, the Benchmark value for each country’s production and consumption of pork is calculated on a scale from 0 to 100.

Concern about farm animal welfare first emerged in Western Europe in the 1960s and later spread to other parts of the world. Initially, it was anticipated that the main driver of improvements would be national animal welfare legislation [1]. Increasingly, however, in Western Europe, the focus shifted to common cross-national legal requirements defined by the European Union (EU) [2,3,4]. A number of EU directives have agreed to the defined minimum standards for national farm animal welfare legislation [5]. Some countries in the EU have implemented legislation going beyond these minimum standards, whereas others have operated at the bare minimum.

However, efforts to improve farm animal welfare by means of regulation at the EU level came to a standstill at the beginning of the new millennium. This development seems to be connected with increasing pressure from global competition in animal production, and also, it would seem, challenges in reaching international agreements on higher animal welfare standards in a considerably expanded EU [4]. Instead, over the past two decades, there has been a growing focus on so-called market-driven animal welfare, where animal products guaranteeing stricter animal welfare requirements than standard products are sold with a special label, typically at a higher price, part of which is used to pay the farmer a premium [6,7]. Thereby, the market both fulfills a demand from a segment of consumers who want products produced with stricter welfare requirements than those delivered by existing legislation and enables farmers to deliver the products (because they are compensated for the extra costs added due to complying with these requirements).

In some countries outside the EU, there have been efforts to regulate farm animal welfare, but with a few notable exceptions, such as Norway and Switzerland, where animal welfare legislation aimed at farm animals has been rather minimal. However, many industrialized countries have seen a growth in market-driven animal welfare since the 1990s.

What has emerged is, therefore, a pluralist approach for addressing farm animal welfare that combines animal welfare legislation with market-driven initiatives [8]. This development has certain advantages [9]. To begin with, it enables countries that are ambitious about animal welfare to be first movers in terms of legislation. Additionally, it is possible for farmers, processors, retailers, and groups of consumers, sometimes supported by state-driven labeling initiatives, to collaborate in setting standards that secure a level of animal welfare challenging to achieve by means of legislation alone [7]. 

However, it is difficult to track how much is actually being achieved in terms of improved animal welfare, particularly when it comes to cross-country comparisons [9]. How, for example, does a country like Sweden, with relatively high legal standards of animal production but a very small market share for market-driven animal welfare, can be compared with the UK, which has a lower legal standard for animal production, but a larger market share for animal products with special welfare labels?

To deal with this problem, the authors of this paper developed a tool for Benchmarking animal welfare initiatives at a national level. Our approach was inspired by comparisons of economic competitiveness between countries—comparisons which, among others, have been used by the World Economic Forum [10]. Like competitiveness, animal welfare is a complex concept, and thus, a tool using economic methods may enable aggregation of seemingly incomparable entities, such as all the aspects of how farm animals are housed and cared for.

We were not the first to attempt to make large scale comparisons of farm animal welfare or animal welfare in general, but our approach differed in important ways from the three other initiatives of which we were aware:

One such initiative is The Business Benchmark on Farm Animal Welfare [11], which aims to benchmark food companies. The Business Benchmark initiative is mainly focused on reporting what companies do in terms of transparency, policy, and commitment rather than on measuring the outcome for the affected animals. However, a section on animal outcomes is included, where the companies involved are asked to report on the proportion of source animals that are housed without close confinement (for example, sows), that are free from routine mutilations (for example, tail docking of piglets), that are subject to pre-slaughter stunning, and that are transported within specified maximum journey times. The companies involved are also asked to report on key welfare outcome indicators. These include negative indicators, for example, tail biting and lameness in pigs, and positive indicators, including the ability to express natural behaviors. A scoring system is linked to the reporting of these outcomes, but what is scored is mainly the degree to which the outcomes are actually reported.

Another initiative is the Animal Protection Index [12]. The methodology for this web-based index was developed by a working group led by the international animal protection NGO, World Animal Protection. The working group also included senior representatives of a number of other animal protection NGOs working internationally, and academic experts in animal welfare were consulted. This tool works at the level of countries; in total, 50 countries are indexed in the latest version from 2020. Based on the index, countries are ranked on a seven-point scale, ranging from A (best) to G (worst). The focus is on legislation and state policy; private initiatives, such as animal welfare labels, are not included. In other respects, the scope is very broad. So, besides the protection of farm animals, the index also includes the protection of laboratory, zoo, hobby, and companion animals (among other things). Animal protection is understood more broadly than welfare (for example, including the euthanasia of surplus animals as a negative). Even though there is a sub-index for farm animals, this covers a range of species and focuses on only a few binary parameters for each species. For example, for pigs, the focus is on whether or not there is a full ban first on sow stalls and farrowing crates and, second, on piglet mutilations.

A third approach is to measure directly the welfare of animals found in a representative sample of farms in different countries. One thoroughly developed approach for doing this can be found in the protocols emerging from the EU funded project Welfare Quality®, which also includes protocols for assessing pig welfare [13]. There have been initiatives, for example, in Denmark [14], to use modified versions of these protocols to assess and compare aspects of welfare at the national level for different farm species, including pigs. However, so far, no representative national assessment of animal welfare, even for one species or one country, has been presented, and there are several reasons to doubt whether this approach will succeed in generating cross-country comparisons of farm animal welfare for pigs and other species. Firstly, there are some theoretical problems regarding the procedures for adding up the different inputs into a meaningful and valid aggregate animal welfare score [15]. Secondly, such an undertaking would require very comprehensive and extremely costly collections of data that are unlikely to occur due to the complexity of the methodology required. Thirdly, we found it unlikely that legislation would be passed that enables random sampling of farms to be assessed.

Our approach, unlike the first two tools above, had a key advantage shared with Welfare Quality®: it was based on a reasonably comprehensive view of the requirements that were likely to affect the experiences of the animal during production and transport. Unlike Welfare Quality®, we did not aim directly to measure the reactions of the animals via so-called animal-based parameters, with the practical difficulties noted above. Instead, we used a proxy in the form of animal welfare experts, who assessed how different aspects of the environment and care provided to the animals would affect their well-being. We focused on on-farm and transport requirements (but, in contrast with *Welfare Quality*®, did not include requirements around slaughter). This approach, in our view, provided a feasible way to get reasonably close to an assessment of the actual welfare of the animals while allowing for comparisons across countries that take into account both legislation and market-driven initiatives to ensure animal welfare.

In the following, we have presented our approach in more detail and have shown how it applies in the case of comparing pig welfare across five European countries. In the discussion, we considered the potential advantages and limitations of our approach compared to the three mentioned alternatives.

## 2. Materials and Methods 

### 2.1. Initial Selection of Countries

Six EU countries were initially selected to serve as cases for developing our tool: Denmark (DK), Germany (D), the Netherlands (NL), Spain (E), Sweden (S), and the United Kingdom (UK). Four of these countries—Denmark, Germany, the Netherlands, and Spain—have relatively large pig production sectors and are all net exporters of pork. Two of them—Denmark and the Netherlands—export a majority of their production, while Germany and Spain serve large home markets. The two remaining countries—Sweden and the UK—have relatively small pig production sectors and are net importers of pork (see Appendix A).

By being members of the EU (the UK only until the end of 2020), all the countries have to comply with the minimum animal welfare requirement regarding pig production defined by the EU. In addition, four of the countries have legislation that goes a little further, while a sixth—Sweden—has legislation that goes much further. The last country, Spain, has legislation that is in line with EU requirements. 

Another difference between the countries concerns market-driven animal welfare. This seems to be highly developed in the UK and the Netherlands, somewhat developed in Denmark and Germany, and less developed in Spain and Sweden. Therefore, the selected countries have an interesting spread when it comes to approaches to dealing with pig welfare.

### 2.2. Identification of Country-Specific Welfare Initiatives

Based on previous work by the authors [6,16,17,18], by use of the network of particularly B.F. and P.S. in the six countries, by support from public webpages (for a list see Appendix A) and literature in the field [19,20,21], we created an overview for each country of major initiatives, aiming to raise pig welfare above the level of most efficient pork production in term of costs incurred per produced unit. 

These initiatives include legal requirements at EU level, national legal requirements that go beyond what is required by the EU, requirements made by publicly- or state-supported animal welfare labels, and requirements made by private animal welfare labels, which are labels managed by companies involved in pig production, by retailers, by animal welfare NGOs, or through collaboration between some of these. 

In Spain, the most important current welfare initiatives, apart from national legislation implementing the EU requirements, are based on the Welfare Quality® protocols. Here, welfare is largely defined not in terms of what is provided to the animals but on animal-based measures of how the animals respond to their conditions. Therefore, Spain ended up being excluded from this study. For a list of the welfare initiatives for the five countries included, see Appendix A.

### 2.3. Definition of Welfare Dimensions

To be able to compare the different initiatives, we defined a number of welfare-dimensions, e.g., provision of space, rooting material, and the like to be used by welfare experts when making assessments. The dimensions were chosen to cover the most important welfare requirements adopted by the different pig welfare initiatives. For pig production, we ended up having 15 dimensions, relating to the keeping of sows, piglets, and slaughter pigs. See Table 1 for an overview of the dimensions.

Within each welfare dimension, a number (ranging from two to seven) of possible gradings were defined. The starting point for each dimension was the minimum found in production systems, which only focused on efficient production. The other gradings reflected what was typically found in welfare initiatives. For example, the provision of space for a sow could range from 3 m^2^ to 10 m^2^ plus outdoor area.

The aim was to ensure that each pig welfare initiative in each of the chosen countries could be defined in terms of a combination of grades within the defined welfare-dimensions.

### 2.4. Calculation of Animal Welfare Score

In order to estimate an animal welfare score in terms of the different grades within the specified animal welfare dimensions we had selected, we used expert assessments. Six recognized university experts, one from each of the countries represented, were asked to estimate the significance, in terms of animal welfare, for each grade within a dimension, on a scale from 0 to 10 (where 0 means that there is not extra welfare compared to the bare minimum, and 10 means that there is the best possible welfare in terms of the relevant dimension). After that, the experts were asked on a scale from 1 to 5 to assess, for each welfare dimension, which weight it should have in giving a full picture of the welfare of the affected pigs. This was done by means of an online questionnaire (the text of the questionnaire can be found in Appendix A).

Based on the answers from the experts, values both for the welfare score of each grade within each welfare dimension and for how much each dimension should contribute to the total welfare were calculated as a weighted sum. We added up the answers of the different experts as a simple mean. We also estimated median values, but as the means and the median values were very similar, we chose to do the further calculations with mean values only. All the responses and both mean and median values can be seen in Appendix A. Furthermore, confidence intervals for the expert scores are presented below as part of the sensitivity analysis in Section 3.3.

In light of the expert responses, it was possible to calculate the contribution of each initiative to pig welfare. This was done on a scale from 0 (=welfare is as it would be if there was only focus on the most efficient production of pork) to 100 (=the best possible welfare achievable within commercial pig production) (see Appendix A).

### 2.5. Weighting According to Proportions of Sows, Piglets, Weaners, and Finishers

Each welfare-dimension was weighted relative to what proportion of pigs are produced in this way in relation to the number of pigs living at each moment in time. Thus, for example, dimensions relating to the welfare of sows only made up a small part of the total population (measured in “lived pig years”, where every year counts the same irrespective of whether it belongs to a sow, a slaughtered pig, a small pig, or a piglet). 

Even though a sow on average lives three to four years and a slaughtered pig only lives for a few months, at every moment of time, many more slaughter pigs than sows exist; therefore, the welfare-dimensions relating to slaughter pigs would have more weight than the welfare-dimensions relating to sows. Besides, the relative share of sows, small pigs, and slaughter pigs vary between the countries because some countries export small pigs to be fattened in another country. Compared to the EU average, Denmark, for example, has a relatively large production of piglets and a relatively small production of fattening pigs. The opposite is true for Germany. It is, therefore, necessary to weight welfare parameters for, for example, piglets higher in Denmark to obtain realistic averages. The coefficients—used for the weights—were calculated by comparing the stock of the three groups—sows, piglets, and fattening pigs, to the average stock in the EU (see Appendix A).

### 2.6. Methodological Assumptions and Considerations

The use of a single benchmark score, based on the sum of scores of individual parameters, builds on the assumption that bad welfare in one dimension can be compensated for with good welfare in another dimension. Having a weight for each dimension allowed us to acknowledge that some dimensions might have a greater impact on animal welfare than others in this compensation process. Similarly, using weights based on lived pig years represents the assumption that good welfare for two pigs for six months has the same contribution to overall welfare as one pig having good welfare for one year. 

Such assumptions are, of course, debatable (and even controversial) but, we should note, such assumptions and comparisons are made all the time—though typically not as transparently as here.

### 2.7. Benchmarking Production and Consumption

To be able to calculate a Benchmark-value for the pig production found within a country, we also needed to consider how large a share of the production was covered by each initiative. We measured this relative to the amount (in tons produced). It would also be possible to measure relative to production value, but this might skew the result towards reporting a too high level of pig welfare since pork produced with extra concern for animal welfare typically would have a higher value than standard pork. 

Besides estimating the Benchmark value for the pork *production* taking place in a country, we also wanted to estimate the Benchmark value for pork *consumption* in a country. These two values can differ, and this can be of relevance for cross-country comparisons of pig welfare. This would, for example, be the case if one country has high welfare standards for pork production but, at the same time, imports large amounts of pork produced under poorer welfare conditions, whereas another country is a net exporter of pork, typically with minimal welfare and, at the same time, produces welfare labeled pork for the home market.

The relevant data for production, import, and export of pork in a country can partly be found in publicly available statistics and partly via contacts with academic experts, civil servants, and representatives of slaughterhouses, pig producers, and retailers in the different countries. Particularly, when it comes to market shares of various private welfare labels, it can be difficult to get information, and here some guestimates have been required. For our estimates, see Appendix A.

In the present paper, we only look at Benchmark values related to a certain point of time. However, based either on historical data or follow up studies over a number of years, it should be possible, based on this method, to assess how levels of animal welfare change over time in different countries.

### 2.8. Assessing the Concurrent Validity of the Benchmark

To be able to assess the concurrent validity of the Benchmark, we compared the Benchmark values of a number of Danish initiatives. This was possible since the authors of this paper already knew these labels well. In Denmark, there are a number of labels set up to allow consumers to buy pork produced with a special focus on improved pig welfare. Of particular interest are two labels with different gradings: One is a governmental animal welfare label, introduced by the Danish Ministry of Environment and Food, initially to cover pork and pork products. This label has three grades or levels with what is considered ascending animal welfare requirements. Another related but slightly different label was set up around the same time by one of the largest Danish retailers, COOP, also from the outset covering pork and pork products. This label has four grades or levels.

## 3. Results

Of the six academic experts in pig welfare whom we asked to contribute to this assessment, after one reminder, five had filled out the questionnaire for assigning values and weights.

Based on their responses and the collected information about the different welfare initiatives in the five countries studied, the animal welfare effect of each initiative was calculated as a value from 0 to 100, and based on information on production, imports, and exports of pork and market shares, Benchmark values for the production and consumption in each country and for different forms of legislation were calculated.

In the following, we first presented results relating to the concurrent validity of the Benchmark in Denmark, and then we presented results concerning the national comparisons of production and consumption Benchmarks, and, finally, we presented results concerning the reliability of expert responses.

### 3.1. Validity of Benchmark

In Figure 1, we showed the Benchmark values of a number of Danish initiatives. The different grades of the governmental label were marked as F1–F3 in Figure 1, whereas the different grades of the COOP label were marked as C1–C4. Other labels in the figure are “DANISH”, which is the industry standard, “private labels”, which covers older labels with a little extra focus on animal welfare, “UK pigs”, a label that among other things ensures that sows are loose housed except in the farrowing unit, “outdoor”, a label ensuring that piglets are produced in outdoor systems, and that slaughter pigs have access to an outdoor run, and “Danish Organic”. There was a large overlap between the two latter labels and F3 and C2–C4. The largest animal protection NGO in Denmark, Dyrenes Beskyttelse, had its own label that, among other things, required outdoor access for all animals. Due to overlaps, the label was not included as a separate label, but it was still covered since it was typically found on products that were here referred to as Danish Organic, F3, and C2–C4. 

The main results coming out of Figure 1, as regards the concurrent validity of the Benchmark, were as follows: 

Firstly, the big picture looked as expected. The Benchmark showed, as expected, a progression of welfare in the two graded labels, F1–F3 and C1–C4. The “UK pigs” were, as expected, in line with the industry standard, reflecting that EU and national legal requirements were closing the gap to this label that originally distinguished itself, for example, by requiring loose housed sows from weaning to one week before expected farrowing. Finally, “outdoor” and “Danish Organic” had Benchmark levels not too far removed from F3 and C2–C4.

Secondly, there were some unexpected deviations when it came to the details. For example, C1 had a lower Benchmark score than F1 and F2, even though the COOP label was developed partly in protest against what was seen as too lenient requirements for F1 and F2. The explanation of this seemed to be that F1 and F2, unlike C1, had an explicit requirement for handling tail-bites that the experts behind the scores and weights for the Benchmark perceived as important, and this was not compensated for by the fact that C1, on some other counts, scored better than F1 and F2. Similarly, “Outdoor” had a lower Benchmark than F3 and C2, although they all required outdoor production. This difference was mainly due to the above-mentioned explicit handling of tail-bites in F3. The same was true for the difference between “Danish Organic” and F3 and C3.

It should be noted that some differences might not reflect differences in how the pigs were actually kept and cared for but rather the degree to which the requirements for keeping the pigs were spelled out. This pointed to a general issue for the Benchmark that it might favor initiatives where management norms were elaborately spelled out, rather than relying on tacit norms of good practice. We have detailed this further in the discussion.

### 3.2. Benchmark of Production and Consumption of Pork in Five Countries

A comparison of the Benchmark score for pig production in the five countries studied can be seen in Figure 2.

There was a major variation between the Benchmark values for pig production in the five countries studied: Germany and Denmark had Benchmark values that were slightly higher than they would have been if all pig production in these countries had just complied with the EU’s minimal requirements. The Benchmark value of Dutch pig production was a little higher than this, and the production in the UK and Sweden had Benchmark values that were much higher.

There were three possible explanations of why some countries had higher Benchmark values than others: (a) that they had more demanding animal welfare legislation than the others, (b) that they had more market-driven pig welfare based on national production, or (c) a combination of (a) and (b).

Using the information collected, we were able to investigate the relative importance of these explanations, in that we could map the hypothetical situation where all production in a country takes place in accordance with national legislation. We did this in Figure 3.

It could be seen by comparing Figure 2 and Figure 3 that the main explanation for why Sweden came out with a high Benchmark value for its pig production was the high welfare requirements upheld in national legislation. Since the Benchmark value for Swedish production was more or less the same as it would have been if all production in Sweden was carried out in accordance with Swedish legislation, Sweden could not have much pig production aimed at products sold with a special welfare label. 

On the other hand, in the UK, the relatively high Benchmark value for local pig production could not be explained by national legislation since the requirements of British animal welfare legislation, as can be seen from Figure 3, were only slightly higher than that demanded by the EU. Instead, there seemed to be two other explanations of the high UK benchmark for pig production that complement each other. Firstly, traditionally, a large part of UK pig production has been based on systems with sows and piglets outdoors. Secondly, the UK has a rather high level of pork products that are sold with welfare labels, and these products are, to a large degree, sourced nationally. 

That Dutch pig production came out with a little higher Benchmark value than that found in Germany and Denmark and seemed to be explained in part by stricter legislation (slightly higher space requirements for slaughter pigs, all sows being loose in the mating unit, and castration only carried out with full anesthesia). As can be seen from Figure 3, legal requirements in the Netherlands go slightly further than they do in Germany and Denmark. This, however, only explained part of the difference. The rest of the difference could, it seemed, be explained by the fact that the Netherlands has a relatively well-developed and large (compared to Denmark) national market for pork and pork products with animal welfare labels, most of which seem to be sourced nationally.

The Benchmark for Danish and German pig production was nearly the same, in line with the fact that the two countries have large pig production sectors, of which the Danish sector, in particular, is very much oriented towards exports of standard pork.

There might be a difference between the Benchmark value achieved for pig production within a country and the Benchmark value for the pork consumed in that country, as can be seen in Figure 4.

This figure shows that countries with a high level of welfare in national production, i.e., Sweden and the UK, had a lower Benchmark value for what was consumed. This reflected relatively large amounts of pork imports for local consumption (around 35% in Sweden and around 60% in the UK (see Appendix A)), typically from countries with a lower level of animal welfare in their pig production, and thereby also lower production costs, allowing for a lower retail price (see Appendix A).

The three countries with a large export-oriented pig sector—Germany, Denmark, and the Netherlands—on the other hand, had higher Benchmark values for their local consumption than that for their local production. The likely explanation of this was that most of the pork exported from these countries is produced according to minimal legal requirements, whereas specialized production for products with animal welfare labels is typically going to the national market. 

German consumption only had a slightly higher score than the production score, but both were higher than the legislative minimum because 20% of the production and 25% of consumption are estimated to be Initiative Tierwohl, which has a little higher animal welfare requirements than legislation (more space and stricter bedding requirements). In Denmark, production almost follows legislation, but the benchmark value for consumption was slightly higher than in Germany due mainly to the fact that around 5% of national consumption is of organic and other forms of outdoor pork production. The benchmark value for consumption was higher in the Netherlands than in Denmark and Germany. The latter reflected the high uptake among Dutch retailers and consumers of the so-called Beter Leven label so that over 70% of the pork consumed in the Netherlands carries that label. Most of that is with the lowest level, a single Beter Leven star, which only distinguishes itself from standard production by not allowing castration of piglets, providing a little extra space to slaughter pigs, and having a maximum of eight hours of transport. 

### 3.3. Sensitivity Analysis of Expert Responses

Values of the grades within a welfare dimension and the weights given to each dimension were determined by assessments made by five university researchers from five different European countries, who were experts in pig welfare. A possible worry could be that the assessment would turn out to be highly subjective, with large individual differences between the researchers, and that the mean values and weight used in this study, therefore, were not representative of anything. We checked the degree to which this was the case in three ways.

Firstly, in Figure 5, we calculated the Benchmark of pig production in one country, Denmark, as it would be if it was based on the values and weights of each expert in turn:

As can be seen from Figure 5, there were some dramatic differences in the values. However, comparative rather than absolute values are what matter in the Benchmark. 

Therefore, in Table 2, we calculated the ordering of country Benchmark scores for the five experts:

We could see from Table 2 that the experts differed in their orderings in two ways. Firstly, they differed about whether welfare is better in Danish or German pig production, which is no surprise, since the welfare levels of the pigs produced in the two countries are very close. The other difference concerned Sweden and the UK, which again is no surprise since the levels of the countries are again very close. In both cases, the differences were due to rather marginal differences in the Benchmark scores. The overall picture was that the values and weights given by the experts were sufficiently close for it to make sense to add them up.

Thirdly, we calculated the 95 % confidence intervals for paired differences in the Benchmark scores for pig production. As scores from the same five experts were used to estimate the Benchmarks for all countries, the observations could not be claimed to be independent, so we used a paired analysis. As we were particularly interested in testing whether the Benchmark for Denmark and Germany differed and whether the Benchmarks for Sweden and the UK differed, we calculated the confidence intervals for differences between Denmark and UK, respectively, and each of the other countries and the EU. We found that the Benchmark score for pig production in Denmark was not statistically significantly different from that of the EU and Germany, whereas it was significantly lower than the Benchmark values for the Netherlands, Sweden, and the UK based on 95% confidence intervals. Similarly, we found that the Benchmark for pig production in the UK was not statistically significantly different from the Swedish Benchmark but higher than the Benchmarks in Denmark, Germany, and the Netherlands. Similar analyses could be carried out for paired differences between the other countries as well as for the Benchmarks for pig consumption in the five countries.

## 4. Discussion

Developing a tool to assess animal welfare in a way that enables comparisons across countries, regions, or other large geographically or politically defined entities for different kinds of animal production is, in our view, an important aim. A pluralistic system for managing animal welfare is in place in most developed countries, where minimal legislation combines with numerous market-driven initiatives. Without a tool like ours, it is very difficult to compare animal welfare requirements across states and regions in a systematic way. Therefore, it is also difficult to hold governments, retailers, organizations producing meat and other animal products, and animal welfare NGOs accountable, or for them to have sufficient information to define meaningful animal welfare targets.

In this section, we have discussed the advantages and limitations of the animal welfare Benchmark and its application to pig production and pork consumption.

### 4.1. Advantages of the Benchmark Approach

In the Introduction, we outlined some potential advantages of the Benchmark approach compared to the three alternative approaches that we were aware of:

Compared to The Business Benchmark on Farm Animal Welfare, our Benchmark enables a comparison of welfare scores across whole countries, and it measures the relative contribution of what is done to improve animal welfare. Compared to the Animal Protection Index, our proposed benchmark is at the same time broader, in that it not only includes animal welfare legislation and other state-driven policies but also private initiatives, and more focused in that it allows a comprehensive assessment of a specific field of animal production and consumption, in this case of pigs.

It should be added that these two approaches each have their own important roles in focusing on what food companies and governments are doing to improve farm animal welfare. So, by pointing to the “advantages” of our Benchmark, we were not claiming that what we were doing was better, but rather that our Benchmark could achieve different things that are also worth achieving.

Compared to the Welfare Quality® approach, our Benchmark, in theory, has a major weakness in that it only measures animal welfare by means of proxies in the form of expert valuations and weightings. On the other hand, as we showed in this paper, we were actually able to make large scale comparisons of animal welfare achievements across countries, and this has so far not been done by means of the Welfare Quality® approach, despite the fact that the protocols for doing so have been publicly available for more than a decade. Furthermore, we developed a simple and transparent way of aggregating the different aspects of animal welfare to a simple numerical value—something that has turned out to be a notorious problem for the Welfare Quality® approach [15].

We thought that the Benchmark approach could easily be extended to include other countries, and we were most happy to share all the developed resources with researchers wanting to do so. Even though it will take some work, it is also relatively easy to apply the same methodology to other forms of animal production. We already did so for chicken meat and would soon share our findings relating to this form of production in a separate paper.

Furthermore, it will be possible to use the approach to document the development of animal welfare over time. This can either be done by means of historical data or by the use of prospective collections of data. 

Finally, we thought that the specific findings relating to pig production and pork consumption based on our Benchmark were of interest in at least two respects:

Firstly, it can be seen that countries that uphold high standards in their national pig production, like Sweden and the UK, do indeed face the problem of having these standards undermined through imports from countries producing pork based on lower standards. On the other hand, it can also be seen that the aggregated Benchmark outcome of pork consumption in the countries is still higher than in countries with lower welfare standards in their national pig production. So, the policies of these countries seem to pay off for the pigs (if not for the local pig farmers).

Secondly, it can be seen that not only animal welfare legislation but also market-driven initiatives make a major difference when it comes to achieving a better Benchmark value for pig welfare. Thus, it is interesting to see the levels of pig welfare achieved in the UK despite the requirements of the national legislation not extending beyond what is found in countries with large export-oriented pig production, such as Denmark and Germany. An interesting case worth looking into more carefully is the Netherlands, which, despite having competitive, export-oriented pig production based on market-driven initiatives, manages to achieve Benchmark values better than those achieved in Denmark and Germany.

### 4.2. Limitations of the Benchmark Approach

There are some obvious uncertainties and limitations of the Benchmark approach:

Firstly, the animal welfare scores and the weights of the different aspects of animal welfare initiatives are subjective, in that they are based on judgments made by specific persons. Even though the persons surveyed in our study are experts in animal welfare, they may still have their biases and idiosyncrasies. It is likely that their disciplinary background influences the relative scores given to different aspects of welfare (Bracke and colleagues [22] found such an influence, but note also that Otten and colleagues [23] only found a minor influence from disciplinary background). Therefore, we compared our expert responses and found that while there are differences in the values ascribed, the experts end up with very similar orderings of the welfare levels found in the countries studied. However, we only used a rather small group of experts.

There are several things that could be done to improve this. Not least, it would be good to redo the valuing and weightings with a much larger sample of experts selected to ensure a variety of disciplinary and professional backgrounds.

Besides, experts might not only have different mean scores but also differ in how they utilize the full range of possible scores. One way to reduce this kind of bias is to use local as opposed to global scaling [24]. Alternatively, we could have used a local scaling technique where, for each criterion, the lowest score used by an expert is rescaled to 0 and the highest score used by the expert for that criterion to 10. This would adjust for experts using the scales differently, but at the same time, it would introduce a risk that small differences in scores are amplified at an expert level.

We chose to use global scaling, and thereby we relied on an implicit assumption that experts use the animal welfare scale in a similar way. As can be seen in Appendix A, the experts all use the whole scale.

Applying the terminology of Monat [24], the weights used for each dimension can be described as importance weights, as they are based on the decision-maker’s perception of how important a given welfare dimension is relative to the other welfare dimensions under consideration—for example, regarding how important space provision is for a finisher pig relative to avoidance of tail docking.

An alternative could be to use weights based on the decision-maker’s perception of how important the difference between grades in a given dimension is (ranging from worst to best grade) relative to the swings in grades for other welfare dimensions. In our case, this would imply that the weight given to space provisions for finishers would take into account the welfare difference between finishers having 0.5 m^2^ (score 0) and finishers having 2 m^2^ and outdoor access (score 10).

We expected that importance weights would be easier for experts to use and that, therefore, using importance rather than swing weights would result in a more homogenous understanding among experts of the task they were asked to perform in assigning weight to each welfare dimension. Therefore, differences between expert scores are addressed directly in the sensitivity analysis rather than through the re-scaling of the scores.

Besides broadening the group of experts involved, it may be beneficial to study the valuing and weightings of consumers and other stakeholders. This may serve to identify possible blind spots in the views of the experts, but it may also serve to document areas of disagreement as to what is seen to matter when it comes to animal welfare (see, for example, [25] for findings regarding such diversity and disagreement).

Secondly, in selecting certain welfare dimensions and grades, we put up a welfare grid that would not fit existing initiatives in every detail. So, choices have here been made that can be discussed. We contributed to this discussion by being completely transparent about the choices. 

A specific shortcoming concerning the welfare grid is that requirements at slaughter were not included. The five countries were, at the time of the study, all members of the EU, and EU legislation regarding slaughter is meant to be harmonized between the five (even though in practice, there may be differences regarding implementation). Besides, most of the initiatives covered do not have specific requirements for slaughter practices. With further developments in this area, we hope to be able to include requirements regarding slaughter in future refinements of the Benchmark.

Thirdly, even where there is legislation for a given welfare dimension, the level of compliance may vary. For example, implementation of the provision of rooting material required by EU rules varies widely [26].

Fourthly, there is uncertainty about much of the information used in setting up the Benchmark. This is indeed the case when it comes to market shares. We were, however, completely transparent about these assumptions, and we invite people who have better information to share it with us.

Fifthly, there is the problem that two initiatives may, in practice, be exactly similar, but that one comes out better because it is better specified. This, again, can be remedied over time by having more details on rules and how they are implemented. Besides, of course, having specified rules ensures that good practices do not erode over time.

Last but not least, there is a problem with what has been termed the management factor: even if two production facilities on paper are completely the same, still the management of the individual farmer or another stock person may make a huge difference in terms of welfare outcomes [27]. Here, of course, outcome-based measures like most of the measures found in Welfare Quality® are clearly superior. Even though some aspects of management may be covered in stated welfare requirements, they will never be fully covered.

As an alternative to using a direct on-farm assessment like that of Welfare Quality® to make cross-country assessments and comparisons of animal welfare outcomes, which, as argued above, is not practically feasible, it has been suggested (e.g., [28,29]) that meat inspection data is used to cover key aspects of animal welfare outcomes. However, previous studies have shown that different sources of meat inspection data may have serious differences in the threshold for recording. A study comparing routine meat inspection data with more thoroughly obtained data at systematic health monitoring revealed moderate or poor correlations between findings of the two kinds of data for lung and heart conditions, respectively [30]. Further, a study investigating the use of meat inspection data in an animal welfare index concluded that the variation in recording practice was considerable and that it was not even “feasible” to adjust for these differences [31]. For these reasons, we found that, currently, there is no meat inspection data available of sufficient quality for use in assessments of animal welfare.

## 5. Conclusions

The Benchmark we presented here could deliver comparisons of the combined effects of animal welfare legislation and market-driven initiatives on the welfare of the affected animals between different countries better than existing alternatives.

In relation to pig production, there seemed to be a real dilemma between maintaining a high volume of national production and having a high Benchmark value for pig welfare in national production. However, we also discovered that countries like Sweden and the UK, who achieve high Benchmark levels in their national production but import large amounts of pork from countries with lower Benchmark levels in their production, still overall did better in terms of their Benchmark level. The results we presented from the Netherlands showed that it is still possible for a country with high export-oriented pig production to reach promising results in terms of their Benchmark values for pig welfare.

It is our hope that the Benchmark approach presented here would get a wide international uptake and that it could, thereby, contribute to the development of farm animal production that progresses towards better animal welfare.

## Figures and Tables

**Figure 1 animals-10-00955-f001:**
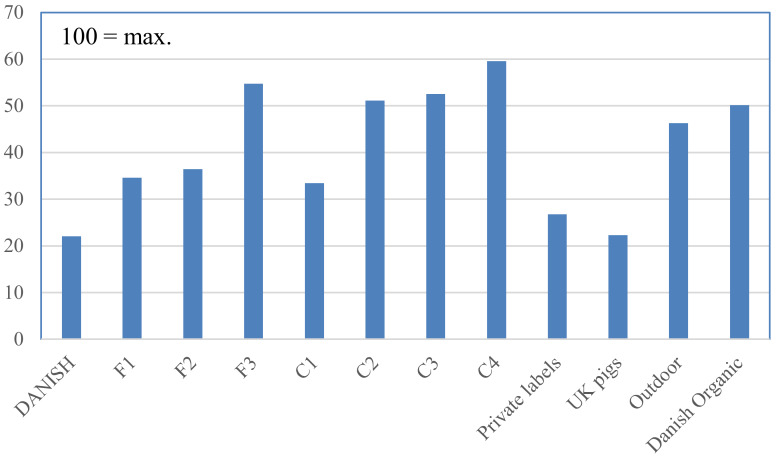
The calculated Benchmark-values (on a scale from 0 to 100) of the different Danish animal welfare initiatives relating to pigs: DANISH is an industry label that from a welfare perspective mainly guarantees compliance with Danish law, F1–F3 are three levels of the official welfare label “Bedre Dyrevelfærd”, C1–C4 are four levels of the retail welfare label “Dyrevelfærdshjertet”, Private labels are some older labels with limited focus on animal welfare, UK pigs complies with the Red Tractor label in the UK, Outdoor is a form of production where sows are housed in outdoor farrowing huts and where finishers have outdoor access, and Danish Organic is a form of outdoor production that complies with the Danish organic standards that are slightly above European Union (EU) requirements for organic production (requiring weaning age of 49 days as compared to the EU requirements of 40 days).

**Figure 2 animals-10-00955-f002:**
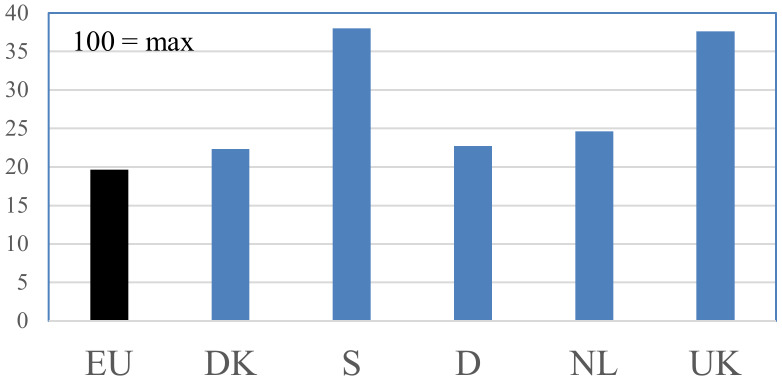
Comparison of aggregated Benchmark scores (on a scale from 0 to 100) for pig production in the five countries studied (both covering production just following national legislation and special, typically labelled, productions with extra welfare requirements). The score for the European Union (EU) represents what the Benchmark score would be if all production was based on the EU’s minimal requirements.

**Figure 3 animals-10-00955-f003:**
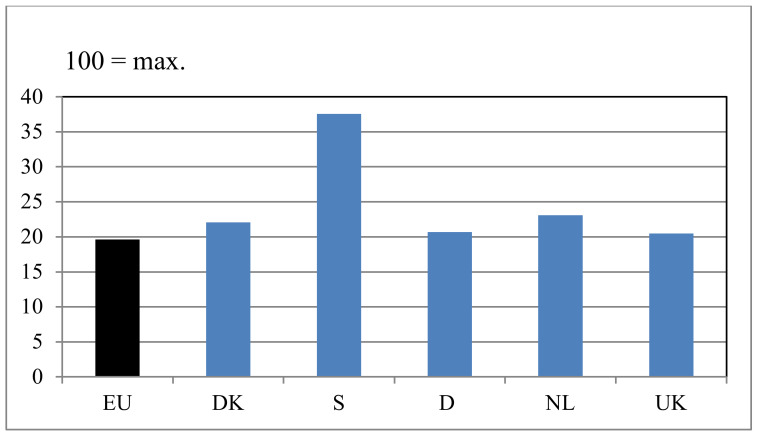
Comparison of Benchmark values (on a scale from 0 to 100) as they would be if all production was done in accordance with minimal requirements from the European Union (EU) (the EU column) or in accordance with national legislation (the other columns).

**Figure 4 animals-10-00955-f004:**
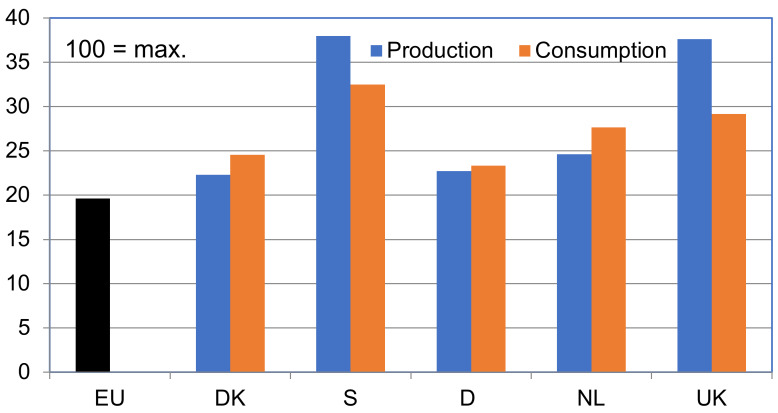
Comparison of aggregated Benchmark scores (on a scale from 0 to 100) for pig production as well as for pork consumption in the five countries studied. The score for the European Union (EU) represents what the Benchmark score would be if all production in a country was based on the EU’s minimal requirements.

**Figure 5 animals-10-00955-f005:**
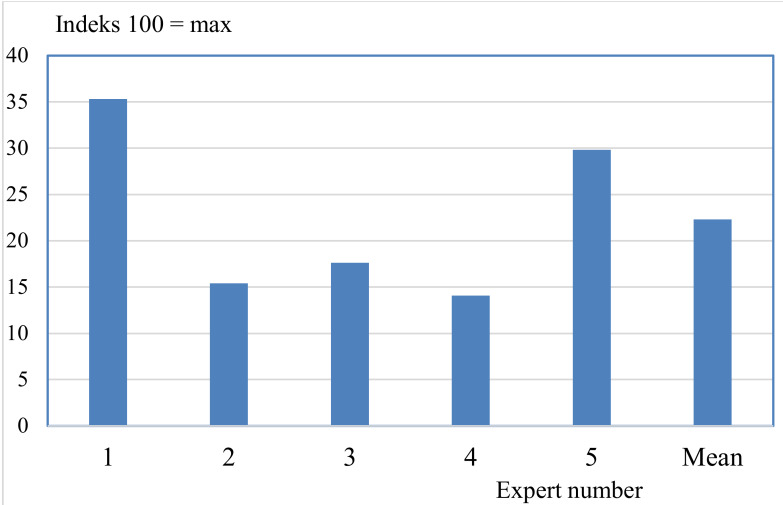
Comparison of aggregated Benchmark scores (on a scale from 0 to 100) for pig production in Denmark calculated for each expert in turn and the mean value.

**Table 1 animals-10-00955-t001:** Overview of questions asked to the experts regarding welfare scores for 15 dimensions (the answering options are explained after the table).

**Sow Welfare** What score (on a scale from 1 to 10) would you give for sow welfare in a production system …
… where the sow is confined for certain periods?
… with the following space requirements for sows with piglets?
… with the following space requirements for sows in group housing?
… with the following access to nest-building material?
… with the following practice for weaning?
**Piglet welfare** What score (on a scale from 1 to 10) would you give for piglet welfare in a production system …
… where the sow is confined for certain periods with different space requirements?
… with the following practice for weaning?
… with the following practice for tail docking?
… with the following practice for surgical castration?
**Weaners and finishers** What score (on a scale from 1 to 10) would you give for the welfare of finisher pigs (100–110 kg) in a production system …
… with the following space requirements?
… with the following restrictions for slatted floors?
**General issues** What score (on a scale from 1 to 10) would you give for the welfare of pigs in a production system …
… with the following practice for bedding?
… with the following practice for investigation and manipulation material?
… with the following practice for keeping animals in pen groups?
… with the following practice for transport to slaughter?

**Table 2 animals-10-00955-t002:** Ordering of Benchmark scores for pig production in the five countries, including a situation where only the minimal European Union requirements apply, based on the values and weight for each expert in turn, and the mean.

Expert/Country	EU	DK	S	D	NL	UK
Expert 1	6	5	1	4	3	2
Expert 2	6	4	2	5	3	1
Expert 3	6	5	1	4	3	2
Expert 4	6	5	1	4	3	1
Expert 5	6	4	2	5	3	2
Mean	6	5	1	4	3	2

1 = best Benchmark score, 6 = worst Benchmark score.

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
