# Peer review of "Benchmarking Farm Animal Welfare—A Novel Tool for Cross-Country Comparison Applied to Pig Production and Pork Consumption"

_animals, 2020, doi:10.3390/ani10060955_

Round 1

Reviewer 1 Report

This well written paper describes a new proposed methodology for the comparison of farm animal welfare in different countries and illustrates its use by comparison of pig welfare in 5 EU member states. This constitutes an interesting study and the paper is of value in the welfare policy debate, in particular because of the production vs. consumption contrasts presented. Whilst the principles of the approach are sound, the paper itself highlights several of the current weaknesses in relation to the data sources and calculations made:

  • The absence of comprehensive data on market shares – this will be difficult to overcomes given commercial sensitivities, but can be further fine-tuned as data allow.
  • Whilst there are published international data on pig populations, I wonder if the paper contains some calculation errors relating to this (see below). It may be that I misunderstand the calculation because it seems not to be fully explained in the text and I cannot access the detailed example in Table S4.
  • The very limited number of experts contributing to the welfare scoring matrix, and their variability in response. I think this is of greater concern, as the outcomes of any comparison are likely to be influenced by the choice of the ‘experts’. It is known that the disciplinary background influences the relative scores given to different aspects of welfare (eg Bracke et al. 2008. Acta Veterinaria Scandinavica, 50: 29.) and this could significantly affect scheme and country comparisons. I think greater discussion of this is merited.
  • The use of resource standards in legislation/labels without consideration of enforcement/adoption of those standards in practice. As discussed in the paper, this is where animal-based outcome measures would be advantageous. The paper suggests that these are impractical at national level. However, there are some food chain data collected by most countries at the abattoir which could be considered (eg Harley et al. 2012. Ir Vet J 65, 11; Tallentire et al. 2018. International Journal of LCA, 24, 1093–1104). Some discussion of utilising this in conjunction with the present approach could be interesting.

Other specific points:

L60. Do they really indicate the actual outcome or the potential outcome? Surely the former depends on enforcement/application.

L85. This development has been discussed in detail in previous studies such as the EU EconWelfare project (eg Keeling et al. 2012. Animal Welfare, 21(S1): 95-105).

l143. If there have been studies for pigs, why not use this reference as the example rather than a dairy one?

l177 Is there a reference for this statement?

L230. Why so few experts? The disciplinary background of these ‘experts’ is surely relevant.

Table S4. I could not see anything on Page 3.

L241 could these data not be handled in a more sophisticated way to produce some confidence intervals in subsequent comparisons?

L253. It is unclear to me how the unit ‘lived pig years’ is derived. If I understand correctly, it is based on the number of animals in any given class which are alive on farm at a specific point in time. However Table S5 does not really enlighten me. A more careful explanation of the calculation of the category weighting would be helpful. How is the ‘coefficient’ given in table S5 calculated and utilised?

Table S5. The calculations here are unclear to me. The numbers in section 1 and section 2 (both ‘000 animals in different categories) do not seem to reconcile. Furthermore, looking at section 2, for countries with little live export you would expect the piglet and fattening pig population at any given time to be at least 10x the sow number. I wonder if there are some calculation errors here - sow numbers for eg Sweden and UK in section 2 appear more than double what I would expect. Maybe I am misunderstanding the calculation being used?

l336. Is there any distinction between total absence of confinement (UK legislation – not yet EU legislation) or no confinement after 4 weeks (EU legislation)

l464. This is a very important section. Could the data variability be used to put ‘confidence intervals’ on the earlier scores?

L571. This could be discussed a bit further – in particular, in relation to the disciplinary background of ‘experts’

L576 I suspect that experts not only have different mean scores, but also differ in how they utilise the full range of possible scores. Would use of a normalisation process for the scores be beneficial?

L577. This could well yield very conflicting conclusions (see eg Averos et al. 2013. Animals: 3, 786-807). Should the final score be based on expert scientific opinion, or societal opinion? Both would have their advocates.

L600. Could the approaches used in this study be supplemented by widely available (though simple) outcome data from eg national abattoir reports?

Reviewer 2 Report

Dear authors, 

this is an interesting and useful paper, worthy of publication after a minor revision, and to which I have only added some comments and suggestions that you may wish to take into account. 

Detailed comments

L 73-74 Possibly worth mentioning the underlying reason(s) for this standstill, which is indeed evident. 

L 90-91 The problem is usually that legislation only sets minimum standards because it is the result of prolonged negotiation by the different lobbying actors. Therefore, it is typically a compromise. When they operate according to legislation, normally farmers have no incentive to introduce higher standards of animal welfare because they receive no financial compensation for the investments (except if the specific member state uses funding from the Common Agricultural Policy to promote higher welfare standards, which is rarely the case). So the only other possible way to go is via market initiatives, whereby the consumer pays a premium price, thus sharing part of the economic burden of giving animals more space, environmental enrichment, outdoor access, etc. These market initiatives pay off because there is consumer demand. In this sense, EU legislation is failing to address citizens' concerns about animal welfare, and the market is compensating by filling a specific market segment. 

L 92-94 Exactly. When you have all these parallel initiative with different standards, benchmarking and comparing is a challenge. 

L 108 ,that -> ,which

L139 directly to measure -> to measure directly 

L 143 Are these initiatives or proposals? The authors later state that they doubt these assessments will ever take place. 

L 150-51 realistically speaking are not going to take place > I would suggest modulating the sentence slightly, along the lines of "are unlikely to occur due to the complexity of the methodology required"

L153 It: it 

L 160-61 Is there a specific reason why slaughter was not included? The events around death arguably matter for pig welfare.

L 187 So -> Therefore (or hence), 

L 272-74 Thanks for the transparency! 

L 316-19 Can be omitted. 

L 331 Was the label of Dyrenes Beskyttelse included in the analysis? If yes, under which category? If not, why? 

L 383, 457 etc. "significant": is this statistically significant? Otherwise please used different terminology

L 390 That> that

L 528-30 I agree, this is a limitation. However, I also agree that it would be challenging to realistically benchmark animal-based outcomes across farming systems and across countries. Another limitation is that you did not consider the slaughter phase (see my previous question). 

L 555 As you have not measured the effects of these provisions on the pigs you cannot confidently say that they "paid off" for them.

L 557-58 I am not sure that you can discuss "outcomes" anywhere as those were not measured. You can only draw conclusions on the criteria/standards (which are more or less high, and more or less important according to the experts), but not on the effects of the provisions on the animals. This is the scope of this study, IMO.

L 563-64 Again, please make sure that you consistently use "criteria" or "standards" but not "outcomes"

L 597 Even > even

L 608-09 Again, please refer to pig welfare standards/criteria, not to "levels of pig welfare". As you rightly state in the discussion, pig welfare is not measured on the animals with your method. One could argue, for instance, that pig welfare is not that much better in the NL (in actual practice) even if the standards are possibly better specified (especially for Beter Leven) or slightly above EU average. 

Round 2

Reviewer 1 Report

The authors have dealt satisfactorily with all the issues raised in my first review. I think this interesting paper is now suitable for publication.

My only minor comment would be the use of a full stop rather than a comma to denote thousands in Annex 7. This night confuse some readers.